# Beyond overconfidence: Embedding curiosity and humility for ethical medical AI

Sebastián Andrés Cajas Ordóñez[1]*, Rowell Castro[2], Leo Anthony Celi[3,4,5], Roben Delos Reyes[6], Justin Engelmann[7], Ari Ercole[8], Almog Hilel[1,3], Mahima Kalla[9], Leo Kinyera[10], Maximin Lange[3,11], Torleif Markussen Lunde[12], Mackenzie J. Meni[13], Anna E. Premo[14], Jana Sedlakova[15]

1 MIT Critical Data, Massachusetts Institute of Technology, Cambridge, Massachusetts, United States of America, 2 MIT School of Engineering, Massachusetts Institute of Technology, Cambridge, Massachusetts, United States of America, 3 Laboratory for Computational Physiology, Massachusetts Institute of Technology, Cambridge, Massachusetts, United States of America, 4 Division of Pulmonary, Critical Care and Sleep Medicine, Beth Israel Deaconess Medical Center, Boston, Massachusetts, United States of America, 5 Department of Biostatistics, Harvard T.H. Chan School of Public Health, Boston, Massachusetts, United States of America, 6 School of Computing and Information Systems, The University of Melbourne, Parkville, Victoria, Australia, 7 University College London, London, United Kingdom, 8 Cambridge University Hospitals NHS Foundation Trust, Cambridge, United Kingdom, 9 Centre for Digital Transformation of Health, Faculty of Medicine, Dentistry and Health Sciences, The University of Melbourne, Carlton, Victoria, Australia, 10 Mbarara University of Science and Technology, Mbarara, Uganda, 11 King's College London, Institute of Psychiatry, Psychology & Neuroscience, London, United Kingdom, 12 University of Bergen, Faculty of Medicine, Bergen, Hordaland, Norway, 13 Florida Institute of Technology, Melbourne, Florida, United States of America, 14 ETH Zurich, Zurich, Switzerland, 15 University of Zurich, Zurich, Switzerland

* sebasmos@mit.edu

## Abstract

Contemporary medical AI systems exhibit a critical vulnerability: they deliver confident predictions without mechanisms to express uncertainty or acknowledge limitations, leading to dangerous overreliance in clinical settings. This paper introduces the BODHI (Bridging, Open, Discerning, Humble, Inquiring) framework, a dual-reflective architecture grounded in two essential epistemic virtues: curiosity and humility, as foundational design principles for healthcare AI. Curiosity drives systems to actively explore diagnostic uncertainty, seek additional information when faced with ambiguous presentations, and recognize when training distributions fail to match clinical reality. Humility provides complementary restraint, enabling uncertainty quantification, boundary recognition, and appropriate deference to human expertise. We demonstrate how these virtues function synergistically in a dynamic feedback loop, preventing both reckless exploration and excessive caution while supporting collaborative clinical decision-making. Drawing from psychological theories of curiosity and cross-species evidence of epistemic humility, we argue that these capacities represent fundamental biological design principles essential for systems operating in high-stakes, uncertain environments. The BODHI framework addresses systemic failures in medical AI deployment, from biased training data to institutional workflow

**Data availability statement:** No datasets were generated or analyzed during the current study. This manuscript presents a theoretical framework for medical AI design based on conceptual analysis and literature synthesis. All supporting literature and references cited in this work are publicly available through their respective publishers and databases.

**Funding:** The author(s) received no specific funding for this work.

**Competing interests:** The authors have declared that no competing interests exist.

pressures, by embedding uncertainty awareness and collaborative restraint into foundational system architecture. Key implementation features include calibrated confidence measures, out-of-distribution detection, curiosity-driven escalation protocols, and transparency mechanisms that adapt to clinical context. Rather than pursuing algorithmic perfection through pure optimization, we advocate for human-AI partnerships that enhance clinical reasoning through mutual accountability and calibrated trust. This approach represents a paradigm shift from overconfident automation toward collaborative systems that embody the wisdom to pause, reflect, and defer when appropriate.

## Author summary

In this work, we explore a simple idea: medical artificial intelligence should not only aim to be accurate, but also to recognise when it might be wrong. Today, many systems speak with great confidence even when faced with situations they do not fully understand. This can mislead clinicians and put patients at risk. We argue that AI used in healthcare should instead show two familiar human qualities: curiosity and humility. Curiosity helps a system look more closely when something seems unusual or unclear. Humility helps it recognise the limits of its knowledge and ask for human guidance when needed. We combine these qualities into what we call the BODHI framework (Bridging, Open, Discerning, Humble, Inquiring), a dual-reflective approach, where AI systems actively seek more information, express their uncertainty, and clearly communicate what they do not know. By doing so, they can support clinicians rather than overshadow them. Our aim is not to replace human judgment, but to build tools that encourage thoughtful, collaborative decision-making. We believe that designing AI in this way can make clinical care safer, fairer, and more aligned with the values that guide good medicine.

## Introduction

At 7:02 a.m., HECTOR (Humble Electronic Clinical Teaching Operations Resource) awakens in Exam Room 4. The medical AI system analyzes a chest X-ray from a 78-year-old patient presenting with fluid retention and wheezing, then delivers its assessment to resident Felix Morales: "Probability of pulmonary edema: 72 percent. Confidence interval: ± 11. Patient history suggests atypical presentation. You might know something I don't." When Felix clicks "Disagree: Show me what you're unsure of", HECTOR highlights a concerning patch in the lower left lobe and confesses: "I was trained mostly on younger patients. I may not be calibrated for septuagenarian lungs during allergy season. But I'd love to learn." This exchange represents a departure from conventional medical AI deployment. HECTOR exhibits what we term BODHI (Bridging, Open, Discerning, Humble, Inquiring), a framework for

dual-reflective intelligence: curiosity that drives active information seeking when facing diagnostic ambiguity. It then pairs it with humility that acknowledges the system's own limitations, while seeking human collaboration. The system does not deliver confident verdicts; it engages in diagnostic conversation, expresses uncertainty quantitatively, and identifies the boundaries of its training distribution.

HECTOR remains largely fictional. Contemporary medical AI systems typically exhibit the opposite behavior: overconfident automation that lacks mechanisms for uncertainty quantification or appropriate deference to human expertise.

Medical history offers parallels where technological confidence overshadowed reflective caution. Over 60,000 Americans were forcibly sterilized under eugenic programs masquerading as scientific progress. In Tuskegee, 400 Black men were deliberately denied effective syphilis treatment as part of a deceptive government study. Thalidomide, marketed as a miracle sedative, caused severe limb malformations in over 10,000 children worldwide [1]. While these tragedies arose from complex intersections of racism, power asymmetries, and systemic injustice, they share critical epistemic dimensions where both curiosity and humility were systematically absent:

Lack of curiosity manifested as failure to question underlying assumptions: eugenic researchers never seriously explored alternative explanations for observed patterns beyond hereditarian frameworks; Tuskegee investigators showed no inter-est in whether their observational paradigm could be ethically justified; thalidomide manufacturers conducted minimal exploration of potential adverse effects before widespread marketing.

Lack of humility was equally pervasive: eugenic scientists claimed certainty about identifying"genetic unfitness" despite profound limitations in their understanding; Tuskegee researchers exhibited moral arrogance in believing their scientific objectives justified denying available treatment, showing no humility about the limits of their authority over others' bodies; pharmaceutical companies marketed thalidomide with unwavering confidence despite incomplete safety data, displaying no restraint about the boundaries of their knowledge regarding teratogenic effects.

In each case, systems maintained overconfident certainty while abandoning both the curiosity to question their own frame-works and the humility to acknowledge fundamental limitations—epistemic, moral, and practical. They lacked mechanisms to recognize when confidence was unwarranted, when additional information was needed, and when deference to affected communities or restraint in action was essential [2].

As AI becomes more integral to clinical care, machine learning continues to deliver impressive advances. Neural networks now demonstrate exponential improvements across diagnostic domains [3–5], rivaling specialists in tasks from dermatological diagnosis to radiological interpretation [6]. Yet a fundamental architectural flaw persists: current AI systems exhibit overconfidence and struggle with uncertainty quantification, particularly when confronting data distributions that differ from their training environments [7,8]. They optimize for performance metrics without developing the reflective capacity to question their own outputs or seek additional information when facing diagnostic complexity.

Extending earlier work [9], we propose that the solution lies not in algorithmic perfection but in embedding the BODHI framework, a dual-reflective architecture within medical AI systems: the dynamic interplay between curiosity and humility. BODHI captures the essential epistemic virtues that medical AI must embody: bridging the gap between algorithmic capability and clinical wisdom, remaining open to new information and alternative hypotheses, exercising discernment in distinguishing confident predictions from uncertain ones, maintaining humility about limitations, and persistently inquiring when diagnostic uncertainty arises. This represents a departure from existing literature, which typically treats uncertainty quantification, active learning, and ethical constraints as separate technical or philosophical concerns. Our contribution is threefold: (1) formally integrating curiosity and humility as complementary architectural principles grounded in both computational theory and cross-species cognitive science; (2) operationalizing their synergistic implementation through context-sensitive activation frameworks; and (3) demonstrating how these virtues must be cultivated in both algorithmic and human dimensions of sociotechnical systems.

Curiosity represents an intrinsic drive to improve predictive understanding and seek structured patterns in complex clinical environments [10,11], manifesting as active information-seeking when faced with diagnostic uncertainty. In machine

learning contexts, curiosity-driven exploration has proven valuable for navigating sparse reward environments and enabling robust transfer learning [12,13].

Humility complements curiosity by acknowledging the fundamental limitations inherent in both artificial and human intelligence, creating mechanisms for uncertainty quantification, error acknowledgment, and appropriate deference to human oversight. Rather than viewing these as constraints on AI capability, we argue that curiosity and humility function as complementary forces in a dual-reflective loop: curiosity drives the system to explore diagnostic possibilities and seek additional information, while humility provides the restraint to recognize when confidence is unwarranted and human collaboration is essential.

The BODHI framework represents a departure from purely optimization-driven medical AI toward systems that can engage in self-questioning, communicate uncertainty effectively, and actively support clinical decision-making through thoughtful collaboration rather than overconfident automation. Human–AI collaborative systems should be designed to foster conditions in which this dual-reflective architecture enhances rather than replaces clinical reasoning [14,15], creating partnerships that leverage both human intuition and machine precision while maintaining awareness of their respective limitations.

## Methods

To develop the BODHI dual-reflective architecture framework, we performed a cross-disciplinary conceptual synthesis integrating three primary domains: 1) computational theories of curiosity-driven learning and intrinsic motivation in artificial intelligence; 2) evolutionary cognitive biology literature regarding epistemic humility and uncertainty quantification in non-human species; and 3) historical analysis of medical ethics failures resulting from overconfidence. We analyzed key historical case studies (Eugenics, Tuskegee, Thalidomide) to identify epistemic deficits, which were then mapped to architectural gaps in contemporary machine learning systems. The proposed framework was iteratively refined through a sociotechnical analysis of clinical workflows to determine necessary intervention points for human-AI collaboration.

## Results

### Beyond optimization: The many faces of hubris

In artificial systems, hubris manifests at both model and deployment levels. At the model level, systems output confident predictions even when wrong, operating without mechanisms to express uncertainty, admit error, or defer to human oversight. Deep neural networks often produce overconfident classifications, misassign high probability to adversarial inputs, and hallucinate content in generative tasks [16–19]. These systems make errors with conviction. This behavior reflects a deeper design failure. Modern AI models are optimized for performance on benchmark metrics, not for self-awareness or caution. They lack calibrated uncertainty estimates or mechanisms for reflecting on their own predictions. In healthcare, this overconfidence becomes contagious. Automation bias leads clinicians to defer to incorrect AI outputs, undermining their judgment. Studies demonstrate that radiologists and ICU staff are prone to follow confident but wrong AI suggestions, resulting in reduced diagnostic accuracy and failures in patient monitoring [20,21]. At the deployment level, hubris emerges when institutions introduce AI into sensitive domains without sufficient evidence of safety or mechanisms for accountability. IBM's Watson for Oncology exemplifies this pattern. Marketed as revolutionary in cancer treatment, it was deployed with limited real-world validation and trained on synthetic cases. It subsequently recommended unsafe treatments and underperformed human oncologists in clinical concordance [22,23]. These failures reveal that the belief that humans can simply "catch" AI errors by staying in the loop is flawed. Human oversight is fallible, especially under time pressure or cognitive load. When AI systems appear competent and confident, users often stop paying close attention, a phenomenon known as automation complacency or automation blindness. Meanwhile, AI systems trained on human-generated data may reinforce the very biases they are meant to correct.

## Curiosity and humility by design - The BODHI framework

The previous section outlined the conceptual case for epistemic virtues in clinical AI. We now turn to their practical realization. This requires more than fine-tuning algorithms. It demands a deliberate shift in how systems are built, deployed, and used. We consider two domains of intervention: AI itself, and the human systems that interpret, implement, and rely on it.

## Defining curiosity and humility for AI systems

AI systems optimized purely for benchmarks often conceal ignorance behind fluent outputs. We propose an alternative: human-AI systems designed from the outset to be aligned with curiosity, humility and reflexivity, virtues that should guide both machines and clinicians. We formalize this approach as the BODHI framework, where each letter captures an essential epistemic virtue: Bridging (connecting algorithmic analysis with clinical wisdom and patient context), Open (remaining receptive to new information, alternative hypotheses, and disconfirming evidence), Discerning (distinguishing high-confidence predictions from uncertain ones requiring additional scrutiny), Humble (acknowledging limitations, quantifying uncertainty, and deferring appropriately), and Inquiring (actively seeking information when facing diagnostic ambiguity). Fig 1 illustrates our dual-reflective trigger framework, demonstrating how these virtues can be operationalized across varying levels of clinical complexity and stakes to achieve more adaptive, transparent, and ethically aligned clinical outcomes. The framework maps four distinct operational modes based on two key dimensions: clinical complexity (ranging from simple to highly atypical presentations) and clinical stakes (from negligible harm to life-threatening/irreversible consequences). As shown in Fig 1, the lower-left quadrant represents routine clinical scenarios where minimal curiosity and modest humility suffice. However, as cases move toward higher complexity or higher stakes, the system must activate increasingly sophisticated reflective mechanisms.

## Curiosity as epistemic drive

Curiosity has proven valuable in addressing persistent challenges in artificial intelligence, including sparse feedback in reinforcement learning [11,24,25], recommendation systems [26–28], and online classification [29,30]. Curiosity-driven learning introduces intrinsic rewards that enable AI systems to learn tasks more efficiently [31]. By encouraging exploration and reducing epistemic uncertainty, curiosity facilitates deeper engagement with the environment, prevents premature convergence to suboptimal solutions, and enhances generalization capabilities [12,26]. In medical AI, we define curiosity as epistemic curiosity—the structured drive to reduce uncertainty and deepen understanding, rather than mere novelty-seeking. Drawing from Berlyne's collative variables theory and Loewenstein's knowledge-gap theory [32,33], curiosity serves as a computational driver that encompasses novelty detection, surprise sensitivity, uncertainty awareness, and responsiveness to incongruent information. Within the BODHI framework, curiosity manifests through the "Open" and "Inquiring" dimensions—remaining receptive to new information while actively seeking clarification when diagnostic uncertainty arises. As illustrated on the right side of Fig 1, curiosity becomes activated when clinical complexity rises, regardless of immediate harm potential. Crucially, epistemic curiosity involves assessing whether an AI system's training data is suitable for the clinical context at hand. As demonstrated in our opening case study, models trained predominantly on younger patients may fail when applied to older populations. A curious system should recognize when its knowledge distribution does not match the presenting problem. Berlyne's theory identifies seven key collative variables that provoke curiosity: novelty (how unfamiliar a stimulus is), change (variation over time), surprisingness (deviation from expectations), incongruity (conflicting information), complexity (difficulty of prediction), uncertainty (outcome unpredictability), and conflict (competing stimuli) [31]. In AI systems, these variables provide a theory-based structure for designing intrinsic reward functions and guiding exploration policies. Computational implementations include: novelty through state visitation counts [34,35], change via differences in learned state representations [36], surprise and incongruity through prediction errors

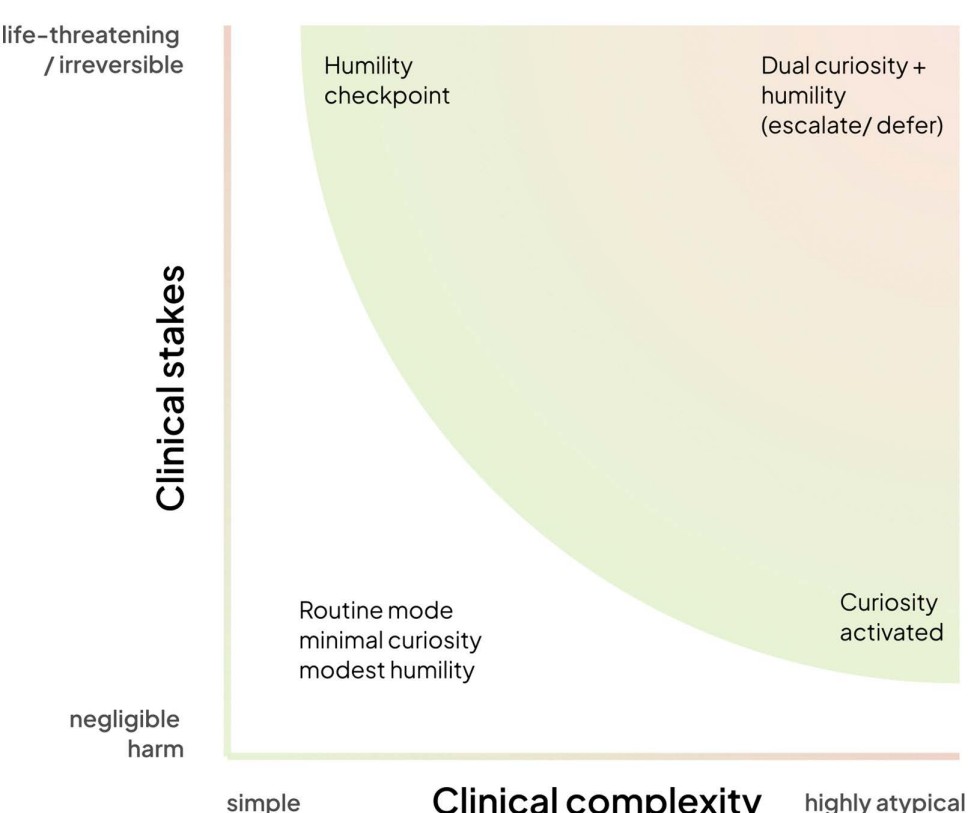

**Fig 1. A rising curved boundary partitions the plane into four zones.** Lower-left - routine mode: negligible harm, simple presentation; minimal prompts. Right - curiosity activated: rising complexity alone triggers information-seeking questions. Upper-left - humility checkpoint: high stakes alone trigger deference to human oversight. Upper-right - dual curiosity + humility: both complexity and stakes are high; the system escalates or defers collaboratively. Shading intensifies toward the upper-right, indicating the growing need for reflective safeguards as cases become both atypical and life-threatening.

[24,37], complexity via progressive task difficulty [38], uncertainty through information gain measures [39], and conflict through contradictory signal detection [28].

### Humility as epistemic restraint

Humility in AI systems represents the capacity to recognize limitations, quantify uncertainty, and defer appropriately to human expertise. Unlike social humility focused on interpersonal dynamics, we emphasize intellectual humility—the metacognitive awareness of knowledge boundaries and willingness to revise beliefs based on evidence [40]. Following Alfano et al.'s framework, we operationalize humility through four dimensions: open-mindedness (versus arrogance), intellectual modesty (versus vanity), corrigibility (versus stubbornness), and active engagement (versus indifference) [41]. Within BODHI, humility is expressed through the "Humble" and "Discerning" dimensions, maintaining awareness of limitations while exercising judgment about when confidence is warranted. As shown in the upper regions of Fig 1, humility

checkpoints are triggered by life-threatening or irreversible scenarios, prompting the system to escalate decisions to human oversight even when the clinical presentation appears straightforward. Humble AI systems implement uncertainty quantification through calibrated confidence measures, recognize inputs that fall outside their training distribution, and maintain mechanisms for appropriate deference to human oversight. We distinguish between appreciative humility that supports collaboration and self-abasing hesitation that undermines clinical utility. The goal is principled restraint: precision when confident, transparency about limitations when uncertain. Humility extends beyond human constructs—emerging evidence suggests it functions as an evolved cognitive strategy across species for navigating uncertainty and fostering cooperation. Bonobos demonstrate prosocial restraint by helping unfamiliar individuals without expectation of reciprocity [42], while rats exhibit epistemic humility by actively seeking additional information when facing uncertain decisions [43]. These findings suggest that humility represents a fundamental biological design principle for systems operating in dynamic, high-stakes environments. For medical AI, this biological perspective reframes humility not as a moral overlay but as a functional necessity. Just as animals evolved to defer or seek help under uncertainty, AI systems deployed in clinical settings must learn to recognize epistemic limits, request oversight, and abstain from decisions when stakes are high and confidence is low.

### Transparency and critical reflection

Transparency mechanisms are essential for operationalizing both curiosity and humility in clinical AI. The "Bridging" dimension of BODHI emphasizes the importance of connecting AI reasoning with clinical understanding through transparent communication. By revealing the internal workings of AI models, these mechanisms empower systems to express their limitations (humility) and identify areas requiring further investigation (curiosity). Explainability methods such as SHAP [44], GradCAM [45], and Integrated Gradients [46] illustrate how models arrive at decisions and pinpoint sources of uncertainty. This intrinsic visibility directly supports humility by making it clear when an AI is less confident or operating near its knowledge boundaries. Furthermore, advanced frameworks like PEEK [47] combine feature attributions with uncertainty quantification in vision-based systems, enabling dynamic transparency that adapts to clinical context. This adaptability fosters curiosity by highlighting novel or ambiguous presentations that deviate from expected patterns, prompting further exploration.

Critical thinking in medical AI involves systematic reflection on reasoning processes, directly supporting both the curious drive to explore and the humble recognition of limitations. Techniques including counterfactual analysis [48], model criticism [49], and frameworks such as ReAct [50] and Reflexion [51] enable models to explore alternative reasoning paths and assess consistency. This capacity for self-assessment is a cornerstone of humility, allowing the AI to question its own outputs. In clinical applications, this translates to presenting differential diagnoses, highlighting contradictory evidence, and explicitly modeling diagnostic uncertainty to support collaborative decision-making between AI systems and healthcare providers. Such mechanisms cultivate curiosity in clinicians by encouraging them to consider multiple hypotheses and engage deeply with the AI's reasoning, rather than passively accepting a single confident prediction.

## Discussion

### Embedding virtue in human-AI systems

Algorithms do not operate in isolation—they function within complex sociotechnical systems where human cognition, institutional pressures, and technological affordances interact dynamically. The successful implementation of BODHI and its dual-reflective AI require deliberate cultivation of curiosity and humility not only in algorithmic design but throughout the entire ecosystem of clinical practice, from individual clinician behavior to organizational culture and regulatory frameworks.

### Fostering clinical curiosity through human-AI collaboration

Human curiosity in clinical practice manifests as a multifaceted cognitive process involving exploration, surprise sensitivity, and the intrinsic motivation to resolve diagnostic ambiguity. In traditional medical education, curiosity is cultivated through

case-based learning, differential diagnosis exercises, and the systematic questioning of clinical assumptions. However, the integration of AI systems creates new opportunities and challenges for maintaining this epistemic virtue.

A curious clinician does not relinquish critical thinking to automated systems but leverages AI as a cognitive amplifier, particularly in routine scenarios where pattern recognition may be compromised by fatigue, time pressure, or cognitive overload. This requires a fundamental shift from viewing AI as an oracle providing definitive answers to understanding it as a sophisticated tool for hypothesis generation and pattern detection that demands active engagement and critical evaluation.

User interface design plays a crucial role in promoting curiosity-driven clinical reasoning. Interfaces that highlight di-agnostic uncertainty through visual cues, probabilistic displays, or confidence intervals invite clinicians to engage more deeply with the reasoning process [52,53]. Systems that flag unusual cases based on statistical outliers, demographic mismatches, or novel symptom combinations can trigger the kind of diagnostic curiosity that leads to more thorough evaluation. Decision aids that present ranked diagnostic alternatives rather than single predictions encourage exploration of multiple hypotheses, while tools that reference similar prior cases or surface contradictory findings from the literature foster broader investigative thinking.

Beyond individual interface design, curiosity-enhancing systems can incorporate active learning mechanisms that identify cases where additional clinical input would be most valuable for both patient care and system improvement. This creates a virtuous cycle where clinician curiosity contributes to model refinement while the model's uncertainty quantification guides clinical attention to the most diagnostically challenging cases. The cultivation of clinical curiosity also requires institutional support through protected time for reflection, access to diverse information sources, and cultures that reward questioning rather than rapid throughput. AI systems can support this by automating routine documentation tasks, thereby freeing cognitive resources for more complex diagnostic reasoning and patient interaction.

## Cultivating clinical humility in the age of AI

Clinical humility begins with the fundamental recognition that diagnostic medicine involves irreducible uncertainty, cognitive limitations, and the ever-present possibility of error. This epistemic stance becomes even more critical in the context of AI-assisted practice, where the apparent sophistication of algorithmic outputs can mask underlying limitations and biases.

Epistemic humility proves particularly crucial for advancing equity and fairness in healthcare, especially within global health contexts where power asymmetries and epistemic injustices systematically shape whose knowledge is recognized, valued, and trusted [54]. Traditional medical hierarchies often privilege certain forms of knowledge—typically Western, academic, and quantitative—while marginalizing patient narratives, community health worker insights, and indigenous healing practices. AI systems trained predominantly on data from high-resource settings risk amplifying these inequities by encoding existing biases into algorithmic recommendations.

A humble approach to clinical AI requires explicit recognition that even highly experienced clinicians make systematic errors due to cognitive biases, knowledge gaps, and contextual constraints. Rather than viewing AI recommendations as either infallible or worthless, humble clinicians develop nuanced approaches to trust calibration that consider case complexity, model uncertainty, patient population characteristics, and the specific clinical context [55,56].

This calibrated trust manifests through several practical behaviors: seeking additional information when AI confidence is low, questioning recommendations that contradict clinical intuition, actively looking for disconfirming evidence, and maintaining awareness of model training scope and limitations. Interfaces that support humble practice display calibrated confidence intervals, communicate uncertainty explicitly through natural language explanations, and provide mechanisms for clinicians to interrogate model logic and assumptions [57].

Humility also extends to interprofessional collaboration, where AI systems can serve as tools for democratizing clinical knowledge and supporting team-based care. By making diagnostic reasoning more transparent and accessible,

well-designed AI systems can empower nurses, pharmacists, and other healthcare professionals to contribute more meaning-fully to patient care while maintaining appropriate professional boundaries.

### Organizational and developmental humility

On the development side, humility manifests through comprehensive transparency and accountability practices that extend far beyond individual algorithmic performance. This begins with rigorous documentation of model training scope, demographic representation, known limitations, and anticipated failure modes. However, true developmental humility requires ongoing commitment to learning from real-world deployment experiences and adapting systems based on clinical feedback.

Model Cards [58] and Dataset Nutrition Labels [59] represent important first steps toward standardizing model documentation, but comprehensive humility requires more dynamic approaches to transparency. This includes real-time monitoring of model performance across different patient populations, systematic collection and analysis of clinician feedback, and proactive identification of emerging failure modes or biases.

Post-deployment humility involves creating robust mechanisms for error tracking, near-miss analysis, and systematic learning from adverse events. Rather than treating model failures as isolated technical problems, a humble approach recognizes them as opportunities for systemic improvement that may require changes in training data, algorithmic architecture, user interface design, or clinical workflows.

Organizational humility extends to procurement and implementation decisions, where healthcare institutions must resist the temptation to deploy AI systems without adequate validation, training, or ongoing oversight. This requires developing institutional capabilities for AI assessment, establishing clear governance frameworks, and maintaining realistic expectations about both the potential benefits and limitations of automated decision support.

### What does it mean for AI to make us better humans?

As AI becomes embedded in more aspects of life, the question is no longer just what can AI do, but what kind of humans do we become through our interactions with AI? Instead of simply optimizing for speed and efficiency, we should ask: can AI systems be designed to foster curiosity and epistemic humility? When interacting with AI systems, can they be designed in a way that supports us in becoming more thoughtful, more aware of our blind spots, and more open to learning? Designing AI systems that are used by humans and yield a new form of human AI collaboration requires thoughtful design that includes normative considerations. These are considerations that concern researchers, designers, but also the public. Answering these normative and ethical questions requires a continuous dialogue with the public and key stakeholders. In general, designing AI systems that create and enable conditions in which humans can self-reflect, recognize their biases (and the biases of AI), gain new perspectives, and acknowledge their strengths and weaknesses is a goal worth pursuing.

This mutual development is particularly critical in clinical practice. Physicians must cultivate complementary virtues to engage productively with BODHI-aligned dual reflective AI: critical discernment to evaluate algorithmic recommendations with calibrated trust, reflexive awareness to monitor their own cognitive biases during AI interaction, and collaborative humility to treat AI as thinking partner rather than oracle. Technical competence alone proves insufficient. Medical education must therefore evolve beyond teaching AI literacy to deliberately cultivating these epistemic virtues through case-based training with AI discordant scenarios, simulation exercises revealing automation bias, and reflective practice protocols. Without such cultivation, even perfectly designed AI systems risk reproducing the overconfidence they aim to prevent.

Humans remain (and should remain) responsible for their own growth: intellectually, morally, and emotionally. AI cannot take over this responsibility. But as we increasingly collaborate with AI, this growth gains new meaning and importance.

As Vallor argues [14], we need to rethink the role of AI so that we can reclaim our humanity and create conditions in which human flourishing is supported. We must define what conditions are needed in human AI collaborations so that key human traits can flourish. Rather than functioning as an all knowing persuasive authority, AI should support our critical thinking and awareness of bias and uncertainty. AI systems should communicate their level of uncertainty and weaknesses, which enables us to know how to collaborate with them and what might be at stake in high risk areas. Human AI collaboration should help us identify and complement each other's strengths and weaknesses, not foster overreliance or false confidence. Implementing values of epistemic humility and curiosity means supporting goals and values beyond mere efficiency; it means connecting with the deeper aims of human activity and professions.

Of course, these values differ across fields. What matters in clinical decision making may differ from what matters in creative work or scientific research. Therefore, the normative foundations of human AI interaction must be defined in dialogue with the specific contexts in which these systems are deployed.

## Conclusion

Medical AI has reached a defining moment. Optimization-driven methods have delivered remarkable diagnostic capabilities, but they have also produced systems that display unhealthy levels of confidence, operate with limited transparency, and fail to align with the uncertainty that clinicians navigate every day. The BODHI framework, Bridging, Open, Discerning, Humble, Inquiring, introduces a dual-reflective architecture that places curiosity and humility at the center of system design, offering a meaningful alternative to the dominant paradigm.

BODHI addresses a critical gap in medical AI deployment by recognizing that failures emerge not from purely technical limitations but from complex interactions between biased training data, institutional pressures, and the fundamental challenge of translating controlled research environments to messy clinical reality. Curiosity drives systems to actively explore diagnostic uncertainty and recognize distribution mismatches, while humility provides essential restraint through uncertainty quantification and collaborative deference. Together, these virtues create AI systems that enhance rather than replace clinical judgment.

The biological foundations of this approach—evidenced in epistemic behaviors across species from rats to bonobos—suggest that curiosity and humility represent fundamental design principles for any system operating in dynamic, high-stakes environments. This perspective reframes these virtues not as moral overlays but as functional necessities for safe, effective medical AI.

Technical implementation requires specific architectural commitments: calibrated uncertainty quantification, robust out-of-distribution detection, curiosity-driven escalation protocols, and adaptive transparency mechanisms. Equally important, the development process itself must embody humility through clear documentation of limitations, post-deployment monitoring, and mechanisms for continuous learning from clinical feedback.

The historical parallels we explored—from eugenic sterilizations to the Tuskegee study—remind us that medical progress without epistemic humility risks perpetuating harm under the guise of advancement. While these tragedies cannot be attributed solely to failures of curiosity and humility, the systematic absence of both virtues enabled catastrophic harm. Without curiosity, these systems never questioned their foundational assumptions or sought contradictory evidence. With-out humility, they proceeded with absolute confidence despite profound uncertainties, claimed authority beyond their legitimate scope, and failed to recognize when restraint and deference were morally essential. Contemporary medical AI risks repeating these patterns when systems lack mechanisms to question their training data (curiosity), recognize distributional limitations (humility about boundaries), quantify and communicate uncertainty (humility about confidence), or defer appropriately to human oversight and affected communities (humility about authority). The BODHI framework aims to prevent such failures by embedding both virtues as foundational architectural requirements.

The future of clinical AI lies not in creating more sophisticated automation but in developing collaborative systems that amplify human capacity for ethical reasoning, nuanced judgment, and compassionate care. This requires fundamentally

redefining success metrics beyond accuracy and efficiency to include measures of appropriate uncertainty, effective human-AI collaboration, and equitable outcomes across diverse populations.

Our vision challenges the field to move beyond the pursuit of algorithmic certainty toward systems that possess the wisdom to pause, the courage to question their own outputs, and the humility to defer when human insight is essential. The measure of medical AI's ultimate success will be found not in its confidence, but in its capacity to support thoughtful, collaborative care that serves human flourishing while remaining perpetually aware of its own limitations.

## Acknowledgments

The authors received no specific funding for this work.

## Author contributions

**Conceptualization:** Sebastián Andrés Cajas Ordóñez, Rowell Castro, Leo Anthony Celi, Roben Delos Reyes, Justin Engelmann, Ari Ercole, Almog Hilel, Mahima Kalla, Leo Kinyera, Maximin Lange, Torleif Markussen Lunde, Mackenzie J Meni, Anna E Premo, Jana Sedlakova.

**Formal analysis:** Sebastián Andrés Cajas Ordóñez, Rowell Castro, Leo Anthony Celi, Roben Delos Reyes, Ari Ercole, Almog Hilel, Mahima Kalla, Leo Kinyera, Maximin Lange, Torleif Markussen Lunde, Mackenzie J Meni, Anna E Premo, Jana Sedlakova.

**Investigation:** Sebastián Andrés Cajas Ordóñez, Rowell Castro, Leo Anthony Celi, Roben Delos Reyes, Ari Ercole, Mahima Kalla, Maximin Lange, Torleif Markussen Lunde, Mackenzie J Meni, Anna E Premo, Jana Sedlakova.

**Methodology:** Sebastián Andrés Cajas Ordóñez, Rowell Castro, Leo Anthony Celi, Roben Delos Reyes, Mahima Kalla, Maximin Lange, Torleif Markussen Lunde, Mackenzie J Meni, Anna E Premo, Jana Sedlakova.

**Project administration:** Sebastián Andrés Cajas Ordóñez, Rowell Castro, Leo Anthony Celi, Roben Delos Reyes, Mahima Kalla, Maximin Lange, Torleif Markussen Lunde, Mackenzie J Meni, Anna E Premo, Jana Sedlakova.

**Resources:** Sebastián Andrés Cajas Ordóñez, Rowell Castro, Leo Anthony Celi, Roben Delos Reyes, Mahima Kalla, Maximin Lange, Torleif Markussen Lunde, Mackenzie J Meni, Anna E Premo, Jana Sedlakova.

**Software:** Sebastián Andrés Cajas Ordóñez, Rowell Castro, Leo Anthony Celi, Roben Delos Reyes, Mahima Kalla, Maximin Lange, Torleif Markussen Lunde, Mackenzie J Meni, Anna E Premo, Jana Sedlakova.

**Supervision:** Sebastián Andrés Cajas Ordóñez, Rowell Castro, Leo Anthony Celi, Roben Delos Reyes, Mahima Kalla, Maximin Lange, Torleif Markussen Lunde, Mackenzie J Meni, Anna E Premo, Jana Sedlakova.

**Validation:** Sebastián Andrés Cajas Ordóñez, Rowell Castro, Leo Anthony Celi, Roben Delos Reyes, Mahima Kalla, Maximin Lange, Torleif Markussen Lunde, Mackenzie J Meni, Anna E Premo, Jana Sedlakova.

**Visualization:** Sebastián Andrés Cajas Ordóñez, Rowell Castro, Leo Anthony Celi, Roben Delos Reyes, Mahima Kalla, Maximin Lange, Torleif Markussen Lunde, Mackenzie J Meni, Anna E Premo, Jana Sedlakova.

**Writing – original draft:** Sebastián Andrés Cajas Ordóñez, Rowell Castro, Leo Anthony Celi, Roben Delos Reyes, Mahima Kalla, Maximin Lange, Torleif Markussen Lunde, Mackenzie J Meni, Anna E Premo, Jana Sedlakova.

**Writing – review & editing:** Sebastián Andrés Cajas Ordóñez, Rowell Castro, Leo Anthony Celi, Roben Delos Reyes, Mahima Kalla, Maximin Lange, Torleif Markussen Lunde, Mackenzie J Meni, Anna E Premo, Jana Sedlakova.

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
