## [Decision Letter · Decision Letter 0]

18 Nov 2025

Response to Reviewers
Revised Manuscript with Track Changes
Manuscript
**Journal Requirements:**

1. We ask that a manuscript source file is provided at Revision. Please upload your manuscript file as a .doc, .docx, .rtf or .tex.

2. Please upload separate figure files in .tif or .eps format. Also, remove the figures from your manuscript file but keep the legends.

3. Please provide an Author Summary. This should appear in your manuscript between the Abstract (if applicable) and the Introduction, and should be 150–200 words long. The aim should be to make your findings accessible to a wide audience that includes both scientists and non-scientists. Sample summaries can be found on our website under Submission Guidelines: 

https://journals.plos.org/digitalhealth/s/submission-guidelines#loc-parts-of-a-submission

**Additional Editor Comments (if provided):**
**Reviewers' Comments:**

**Comments to the Author**

1. Does this manuscript meet PLOS Digital Health’s publication criteria?

Reviewer #1: Yes

Reviewer #2: Yes

Reviewer #3: Yes

2. Has the statistical analysis been performed appropriately and rigorously?

Reviewer #1: N/A

Reviewer #2: N/A

Reviewer #3: N/A

3. Have the authors made all data underlying the findings in their manuscript fully available (please refer to the Data Availability Statement at the start of the manuscript PDF file)?

Reviewer #1: Yes

Reviewer #2: Yes

Reviewer #3: Yes

4. Is the manuscript presented in an intelligible fashion and written in standard English?

Reviewer #1: Yes

Reviewer #2: Yes

Reviewer #3: Yes

Reviewer #1: Thank you for the opportunity to review this manuscript- I thoroughly enjoyed reading it. It is a thought-provoking and persuasive piece. The conclusions are well-supported by a range of literature and it provides essential insights for healthcare practitioners, technology specialists and healthcare system managers and policy makers. Well done!

Reviewer #2: This is a highly interesting and valuable paper that will contribute to improving the quality of future medical care. However, there are sections that require additional explanation and elaboration, so I would appreciate the authors’ addressing these points.

1. In the Introduction, the authors mention forced castration and the Tuskegee syphilis study. Further explanation is needed regarding the connection between these two historical cases and the virtues of curiosity and humility, which are the central themes of this paper. These historical events cannot be explained solely by a lack of curiosity and humility; I believe that numerous other factors contributed to triggering these incidents. Since the conclusion also references these events, relevant explanation should be added to one of these sections.

2. This paper centers on discussing AI equipped with curiosity and humility, detailing how to endow AI with these two virtues. However, it lacks discussion on the virtues required of the physicians using such AI. To maximize patient care benefits, the paper should add discussion on what virtues are necessary for physicians using AI, and further, how to cultivate these necessary virtues in physicians.

3. Finally, the current description of Author Contributions is extremely inadequate. It is necessary to clearly state how each of the fourteen authors contributed to the creation of this paper.

Reviewer #3: Thank you for your valuable article. It is suggested that the authors explain more about how the idea for this article was formed and which part of the article adds to the existing literature in the content.

**Do you want your identity to be public for this peer review?** For information about this choice, including consent withdrawal, please see our Privacy Policy

Reviewer #1: No

Reviewer #2: No

Reviewer #3: No

**Figure resubmission:**

**Reproducibility:** To enhance the reproducibility of your results, we recommend that authors of applicable studies deposit laboratory protocols in protocols.io, where a protocol can be assigned its own identifier (DOI) such that it can be cited independently in the future. Additionally, PLOS ONE offers an option to publish peer-reviewed clinical study protocols. Read more information on sharing protocols at https://plos.org/protocols?utm_medium=editorial-email&utm_source=authorletters&utm_campaign=protocols

---

## [Decision Letter · Decision Letter 1]

4 Dec 2025

Beyond Overconfidence: Embedding Curiosity and Humility for Ethical Medical AI

PDIG-D-25-00663R1

Dear Cajas Ordóñez,

We are pleased to inform you that your manuscript 'Beyond Overconfidence: Embedding Curiosity and Humility for Ethical Medical AI' has been provisionally accepted for publication in PLOS Digital Health.

Best regards,

Hadi Ghasemi

Academic Editor

PLOS Digital Health

**Additional Editor Comments (if provided):**

**Reviewer Comments (if any, and for reference):**

Reviewer's Responses to Questions

**Comments to the Author**

Reviewer #2: All comments have been addressed

publication criteria?

Reviewer #2: Yes

3. Has the statistical analysis been performed appropriately and rigorously?

Reviewer #2: Yes

4. Have the authors made all data underlying the findings in their manuscript fully available (please refer to the Data Availability Statement at the start of the manuscript PDF file)?

Reviewer #2: Yes

5. Is the manuscript presented in an intelligible fashion and written in standard English?

Reviewer #2: Yes

Reviewer #2: I have no further comments. All comments have been addressed very well.

**Do you want your identity to be public for this peer review?** For information about this choice, including consent withdrawal, please see our Privacy Policy

Reviewer #2: No
